# Effect of Aerobic Exercise on Intestinal Microbiota with Amino Acids and Short-Chain Fatty Acids in Methamphetamine-Induced Mice

**DOI:** 10.3390/metabo13030361

**Published:** 2023-02-28

**Authors:** Xin Liang, Xue Li, Yu Jin, Yi Wang, Changling Wei, Zhicheng Zhu

**Affiliations:** 1School of Sports Medicine and Health, Chengdu Sport University, Chengdu 610041, China; 2Department of Epidemiology and Population Health, Albert Einstein College of Medicine, Bronx, NY 10461, USA

**Keywords:** methamphetamine, aerobic exercise, microbiota, amino acids, SCFAs

## Abstract

This study aimed to investigate the changes in intestinal homeostasis and metabolism in mice after methamphetamine (MA) administration and exercise intervention. In this study, male C57BL/B6J mice were selected to establish a model of methamphetamine-induced addiction, and the gut microbiota composition, short-chain fatty acids (SCFAs), and amino acid levels were assessed by 16S rRNA, liquid chromatography–tandem mass spectrometry, and gas chromatography–tandem mass spectrometry, respectively. The results showed that 23 dominant microbiota, 12 amino acids, and 1 SCFA were remarkably higher and 9 amino acids and 6 SCFAs were remarkably lower in the exercise model group than in the control group. Among the top 10 markers with opposite trends between the exercise intervention group and model group, the differential microbiomes included *Oscillibacter*, *Alloprevotella*, *Colidextribacter*, *Faecalibaculum*, *Uncultured*, *Muribaculaceae*, and *Negativibacillus*; amino acids included proline; and SCFAs included isovaleric acid and pentanoic acid. Proline was negatively correlated with *Negativibacillus* and positively correlated with pentanoic acid. The results suggested that moderate-intensity aerobic exercise may modulate changes in the composition of the gut microbiota and the levels of amino acids and SCFAs induced by MA administration.

## 1. Introduction

According to the 2022 China Drug Situation Report [1], methamphetamine (MA) abuse leads to structural changes and functional impairment of the cerebral cortex, which in turn leads to executive impairment of higher functions such as cognition, memory, and emotion [2]. When MA intake is forcibly withdrawn, the body’s sensitized reward system does not receive sufficient stimulus signals, and the sense of pleasure fades, resulting in stimulus problems such as seizures, coma, cramps, hallucinations, and delusions [3]. These symptoms are usually accompanied by mood disorders, such as depression, anxiety, and impaired neurocognitive function [4]. In severe cases, organ damage, mental illness, and even homicide or other violent behavior may occur [5]. These symptoms also act as the basis for limiting methamphetamine use disorder (MUD) [6], leading to relapse and creating a vicious cycle [7,8]. Therefore, understanding the mechanisms by which MA induces structural and functional brain damages is important to find effective ways to ameliorate it.

The effectiveness of drug treatments related to MUD is not obvious at this stage, and effective and safe drug treatments with universal applicability remain unavailable [9,10,11,12]. Thus, non-pharmacological treatments have been developed. Exercise as a non-invasive treatment is popular with most patients for its effectiveness and low adverse effects, and it is gradually being used in treating drug addiction [13]. The most popular non-invasive treatment is aerobic exercise. Studies showed that 8 weeks of aerobic exercise can improve the executive and autonomic nervous system function of patients with MUD by enhancing the availability of striatal D2/3 receptors, therefore effectively alleviating symptoms of depression and anxiety [14,15]. The reduction of various oxidative enzymes in the serum of patients with MUD after 12 weeks of aerobic exercise improves cognition and working memory [16]. Short-term moderate-intensity aerobic exercise can suppress cue cravings and enhance the appetite of patients with MUD [17,18,19] Most existing studies focused on the mechanism by which exercise modulates the function of the circulatory system or central nervous system in MUD, but the mechanism by which exercise improves the MA-induced microbiota–gut–brain axis, which is a multi-system combination of the digestive and endocrine systems, needs further investigation [20].

The impact of the “microbiota–gut–brain axis” on the brain has been extensively demonstrated, and disruptions in this system are highly correlated with various neurological disorders and altered behavioral abilities [21], which has led many scholars in the field of MA addiction to investigate the impact of gut microbiota on MUD and brain function [20]. MA administration resulted in changes in the composition of the gut microbiome [21,22,23], which increased the abundance of pathogenic bacteria, decreased the abundance of beneficial bacteria, and caused neurotoxicity in mice [24]. Moreover, MA administration caused gut microbiota dysbiosis and depression-like behavior in rodents after 7 days of MA discontinuation [25]. These results imply that MA administration may have effects on the homeostasis of the gut microbiota, which consequently leads to structural and functional brain damage in the organism. In addition, short-chain fatty acids (SCFAs) are produced by the degradation of amino acids by intestinal microbiota and are an important bridge between microbiota and the central nervous system [26]. SCFAs can enter the blood circulation and exert direct or indirect effects on the organism [27,28,29,30]. Valproic acid can cause neuroadaptation at glutamatergic synapses by attenuating MA-induced reductions in H4K16Ac recruitment on *AMPAR* gene sequences and downregulating striatal glutamate receptors, leading to addiction or related cognitive decline in animals and humans [31,32]. In addition, SCFA supplementation could reduce parA bacteria and increase lactobacilli in the intestine of MA-induced mice, which consequently altered the behavioral abilities of these mice [33].

Exercise is an important factor affecting the microbiota–gut–brain axis [34]. It can directly alter the composition and functional metabolic levels of gut microbiota, stabilize disorders of gut microbiota, interact with amino acids, and regulate SCFA levels [35,36] to reduce neuroinflammation and maintain neuroplasticity [37]. Endurance and resistance training can protect the central nervous system’s autoimmune function by reducing the thick-walled phylum/bacteroid ratio and intestinal mucosal permeability [38]. Aerobic exercise promotes physical function and regulates changes in beneficial bacteria in the brain [39]. Different exercise intensities can have different effects. Medium-intensity exercise can synergize with creatine, which consists of arginine, methionine, and glycine, to improve the activity of antioxidant enzymes, eliminate reactive oxygen species and reactive nitrogen species, and reduce oxidative stress and neuroinflammatory responses in brain neurons [40,41,42]. Low-intensity exercise reduces intestinal ecological dysregulation in mice, increases valproic acid content, and improves postoperative neuroplasticity and cognitive function in mice [37]. In addition, extreme exercise improves overall body health by increasing the levels of SCFAs, such as acetate, propionate, and butyrate [43]. Aerobic exercise increases butyrate-producing taxa, and fecal acetate and butyrate concentrations downregulate pro-inflammatory cytokines and upregulate anti-inflammatory cytokines and antioxidant enzymes [34]. However, few studies have linked exercise to MUD and gut microbiota. The mechanism by which exercise affects gut microbiota, amino acids, and SCFAs in patients with MUD remains to be explored in depth.

In this study, we combined 16S RNA sequencing and untargeted metabolomics approaches by constructing an MA-induced addiction model in mice. We worked to elucidate the effects of aerobic exercise on MA-induced changes in gut microbiota, amino acids, and SCFAs in mice, which could lay the foundation for subsequent studies.

## 2. Methods and Materials

### 2.1. Experimental Subjects

Eighteen 2-month-old C57BL/B6J male specific-pathogen-free mice weighing 20 ± 1 g were purchased from Chengdu Dashuo Experimental Animal Co. and housed in the animal laboratory of the Sichuan Provincial Key Laboratory of Sports Medicine (specific pathogen-free grade) of Chengdu Institute of Physical Education. The mice were housed under the following conditions: 3 mice per cage, 12 h light/dark cycle, 22 ± 2 °C temperature, 52% ± 2% humidity, and free access to food and water. The study was approved by the Ethics Committee of the Chengdu Sports Institute (No. {2022} 56). The whole process was performed according to the NIH guidelines for animal research to ensure that the entire experiment met ethical requirements.

### 2.2. Experimental Protocol

After 1 week of adaptive feeding, all mice were randomly divided into three groups: the saline control group (group C, *n* = 6), the MA model control group (group Ma, *n* = 6), and the MA model exercise intervention group (group Ea, *n* = 6). We considered the effect of feeding and coprophagy on colonic microbiota; therefore, we divided each group of mice into two cages and kept three mice in each cage. MA drug injection (intraperitoneal injection) was started for mice in the Ma and Ea groups as previously described [44]. MA drug injection (intraperitoneal injection) was administered at 1 mg/kg (dissolved in saline at 1 mg/mL) in the Ma and Ea groups for 7 days, and the conditioned place preference (CPP) validation experiment was conducted (drugs were provided by Sichuan Police College–Intelligent Policing Sichuan Provincial Key Laboratory). The mice in group C were injected with 0.9% saline at the same time and in equal doses. MA dose selection was based on prior studies, and the 1-week dosing period chosen for the chronic regimen represents the average number of days/months of exposure observed in human chronic studies [44]. The total 1-week dose was chosen to fit the total daily/monthly use model in humans [45]. In addition, after a multi-literature review, we determined that 1 mg/kg/day MA is a more desirable and common dose to induce drug-seeking behavior in mice [46]. After the CPP was finished, the exercise intervention was started for the mice in group Ea, and the aerobic exercise method used was running platform exercise (SA101, Saionce, nanjing, Jiangsu, China) as previously described [47] with some adjustments. The slope of the running platform was set to 0° and increased at 8 m/min to 12 m/min (3 days) and finally at 12 m/min for 60 min for 11 days for 2 weeks [47,48]. The time for running table practice was from 19:00 to 20:00. During this period, the mice in groups C and Ma were kept in cages for 2 weeks without any intervention. Sampling was performed on the day after the exercise intervention. The experimental mice were anesthetized with 0.5% sodium pentobarbital, and cecum contents were collected into lyophilized tubes, immediately placed in a liquid nitrogen tank, frozen, and stored in a −80 °C refrigerator for molecular biological assays.

### 2.3. Conditional Place Preference Experiments

CPP was used to observe mouse activity in the companion and non-companion boxes for the assessment of psychotropic drug dependence in mice [49]. The mice were injected with MA at the same time as the CPP test. The right box (white wall) was set as the companion box, and the left box was the non-companion box (black striped wall). The experimental procedure consisted of 1-day of habituation and 1-day of baseline testing. In the 1-day habituation, the partitions were removed, and the mice were placed in the central corridor and allowed to move freely between the two compartments for 10 min. In the 1-day baseline testing, the partitions were removed, the mice were allowed to move freely for 10 min, and the residence time in each box was recorded. The experimental equipment for the 7-day conditioned effect and 1-day preference test was provided by Jiangsu Science (SA213). In the 7-day conditioned effect, the experimental animals were injected with MA, placed in the companion box, and removed 30 min later. After an interval of 8 h on the same day, the experimental animals were injected with saline and placed in the non-companion box. In the 1-day preference test, the drug injection was stopped, the partitions were removed, and the mice were placed in the central corridor and allowed to move freely between the two compartments for 10 min. The experimental equipment was provided by Jiangsu Scions (SA213). The remarkable increase in the duration of the test phase compared with the baseline phase indicated successful modeling.

### 2.4. Sequencing of 16SrRNAs

DNA extraction and amplification: The contents of the mouse cecum were collected as samples, immediately frozen on dry ice after collection, and stored at −80 °C. The samples were crushed with a mortar and pestle, and bacterial DNA was isolated from the cecum contents using a DNeasy PowerSoil Kit (Qiagen, Hilden, Germany) according to the manufacturer’s instructions. DNA concentration and integrity were measured by spectrophotometry (NanoDrop 2000, Thermo Fisher Scientific, Waltham, MA, USA) and agarose gel electrophoresis, respectively. PCR amplification of the V3–V4 hypervariable region of the bacterial 16S rRNA gene was performed in 25 μL of the reaction solution. Universal primer pairs (343F: 5′-TACGGRAGGCAGCAG-3′ and 798R: 5′-AGTATCTAATCCT-3′) were used. The reverse primers contained a sample barcode, and both primers were connected to the Illumina sequencing adapters.

Library construction and sequencing: Amplicon quality was visualized using agarose gel electrophoresis. The PCR products were purified with AMPure XP beads (Agencourt, Beckman Coulter), amplified by another round of PCR, and purified with AMPure XP beads again. The final amplicon was quantified using a Qubit dsDNA Assay Kit (Thermo Fisher Scientific, Waltham, MA, USA). Sequencing was performed on an Illumina NovaSeq 6000 with 250 bp paired-end reads. (Illumina Inc., San Diego, CA, USA; OE Biotech Company, Shanghai, China).

The raw data were stored in FASTQ format, and the primer sequences in the raw data were cut off using cutadapt software. Then, the qualified double-ended raw data from the previous step were subjected to quality filtering, noise reduction, splicing, and chimera removal using DADA2 with the default parameters of QIIME 2 (2020.11) to obtain representative sequences and amplicon sequence variant (ASV) abundance tables. The representative sequences of each ASV were selected using the QIIME 2 software package, and all representative sequences were annotated against the database. The Silva (version 138) database was used for comparison. Species alignment annotations were analyzed using the default parameters of the q2-feature-classifier software.

### 2.5. Amino Acid Identification

Amino acids were detected using gas chromatography–tandem mass spectrometry (MS/MS) as follows. Cecal content samples (30 mg) were placed in a 1.5 mL EP tube pre-chilled at −20 °C, added to 400 μL of methanol–water (4:1 *v*/*v*, containing 0.1% formic acid with internal standard succinic acid-2,2,3,3-d4), and two small steel beads, refrigerated at −20 °C for 2 min, and then grounded (60 Hz, 2 min). Each sample was sonicated in an ice bath for 10 min, left to stand at −20 °C for 30 min, and centrifuged at 12,000 rpm for 10 min at 4 °C. The supernatant (300 μL) was collected. Methanol–water (300 μL, 4:1 *v*/*v*, containing 0.1% formic acid with an internal standard of succinic acid-2,2,3,3-d4). The sample was vortexed for 30 s, sonicated for 5 min in an ice bath, and then centrifuged at 12,000 rpm for 10 min at 4 °C. The supernatant (300 μL) was collected, combined with the other supernatant for a total volume of 600 μL, and then vortexed. Subsequently, 100 μL of the supernatant was evaporated in a wide-lined tube. An 80 μL solution (15 mg/mL) of methoxamine hydrochloride pyridine was added to the glass derivatization vial, vortexed, and then shaken for 2 min. The oxime reaction was performed in a shaking incubator at 37 °C (60 min). The samples were removed and then stored at 20–25 °C for 30 min until analysis.

The target metabolites were qualitatively and quantitatively determined using gas chromatography (Thermo Scientific TSQ9000). The chromatographic conditions were as follows: DB-5MS capillary column (30 m × 0.25 mm × 0.25 μm, Agilent J&W Scientific, Folsom, CA, USA), high-purity helium (purity not less than 99.999%), a flow rate of 1.2 mL/min, and an inlet temperature of 260 °C. The injection volume was 1 μL with no split injection, and the solvent delay was 4 min. The programmed temperature rise was as follows. The initial temperature of the column temperature chamber was 50 °C for 0.5 min, and the programmed temperature rise was 15 °C/min to 125 °C for 2 min, 8 °C/min to 210 °C for 2 min, 11 °C/min to 270 °C for 1 min, and 25 °C/min to 305 °C for 3 min. Mass spectrometry was performed under the following conditions: an electron bombardment ion source (EI), an ion source temperature of 300 °C, a transmission line temperature of 280 °C, a scan mode of a selected reaction detection scan, and a mass scan range of *m*/*z* 40–600.

### 2.6. Identification of SCFAs

The SCFAs were detected using liquid chromatography–tandem mass spectrometry with the following steps. About 30 mg of cecal contents were collected, added to 300 µL of 50% acetonitrile–water solution (*v*/*v*, containing an internal standard mix {2H9}-pentanoic acid and {2H9}-hexanoic acid pre-cooled at 4 °C), ground for 3 min (pre-chilled at −20 °C), and extracted by sonication in an ice water bath for 10 min. The extract was centrifuged at 12,000 rpm for 10 min at 4 °C, and the supernatant was diluted 5 times with 50% acetonitrile–water solution (*v*/*v*). Then, 80 μL of the diluted supernatant was transferred to the injection vial. The injection vial containing the extract was added to 40 μL of 200 mM 3-NPH (50% acetonitrile–water solution, *v*/*v*). The injection glass vial containing the extraction solution was added with 40 μL of 120 mM EDC–6% pyridine (50% acetonitrile, *v*/*v*) and allowed to react for 30 min at 40 °C. The samples were cooled on ice for 1 min. Then, 160 μL of the supernatant was aspirated with a syringe, filtered through a 0.22 μm organic-phase pinhole filter, transferred to a brown injection vial, and then stored at −80 °C until analysis on the machine.

In this experiment, the target metabolites were qualitatively and quantitatively determined using UPLC–ESI–MS/MS. The chromatographic conditions were as follows: injection volume of 5 μL, flow rate of 0.35 mL/min, and mobile phases of A (0.1% formic acid–water solution) and B (acetonitrile). The gradient elution procedures were as follows: 0 min A/B (90:10, *v*/*v*), 1 min A/B (90:10, *v*/*v*), 2 min A/B (75:25, *v*/*v*), 6 min A/B (65:35, *v*/*v*), 6.5 min A/B (5:95, *v*/*v*), 7.8 min A/B (5:95, *v*/*v*), 7.81 min A/B (90:10, *v*/*v*), and 8.5 min A/B (90:10, *v*/*v*). The MS conditions were as follows: curtain gas of 35 psi, collision-activated dissociation parameter of the medium, positive ion spray voltage of 5500 V, negative ion spray voltage of −4500 V, ion source temperature of 450 °C, column temperature of 40 °C, spray gas (Gas1) of 50 psi, and auxiliary heating gas (Gas2) of 60 psi.

### 2.7. Statistical Analysis

The behavioral data were statistically processed using SPSS 26.0 software and graphically plotted using GraphPad Prism 8.4.2. Results are expressed as *x* ± *s*. Normal distributions between two or three groups were determined using an independent sample *t*-test and a one-way ANOVA, respectively. Non-normal distributions between two or three groups were determined using the Mann–Whitney U test and the Kruskal–Wallis test, respectively. Statistical significance was considered at *p* < 0.05.

The analysis software used for the analysis of 16S rRNA was QIME2 (2020.11) and R (v 3.2.0). Principal coordinate analysis (PCoA), heat map (default parameters were used for heat map plotting, and simultaneous clustering of colonies and samples was performed), linear discriminant analysis (LDA, score > 3), and LDA effect size (LEfSe) analysis were performed using R. The differences among the three groups were calculated by ANOVA. Alpha diversity was assessed using Shannon, Simpson, phylogenetic diversity, and chaos. The distance matrix between samples (beta diversity) was assessed using Bray–Curtis PCoA. We selected LEfSe to identify the dominant bacterial taxa in mice before and after the exercise intervention. Then, the key distinguishable ASVs were found by a random forest algorithm to construct ASV taxonomic distributions and calculate the relative abundance of different levels of gut microbiota.

Subsequently, we used the HIPLOT platform [50]. D2 clustering, and Pearson correlation calculations for the heat mapping of data on amino acids and SCFAs and determining the correlations with the gut microbiome. Random forest algorithms (MeanDecreaseGini was used by default for graphing) on the Oebiotech cloud platform [51] were used to find the correlations between exercise and MA addiction.

## 3. Results

### 3.1. MA-Induced Conditioned Place Preference in Mice Exhibited Addictive Behavior

Figure 1 demonstrates the addictive behavior of mice after MA administration. The results showed that after the administration of the drug, the active time of the mice in the Ma and Ea groups was significantly higher in the right box compared to the baseline level (Figure 1A,B), while the active time in the left box was significantly reduced in both groups (Figure 1C,D), which was significantly different from the baseline level (*p* = 0.0005, *p* = 0.0114). At the baseline level, no significant difference (*p* > 0.05) was found between the two groups of mice in the right box (Figure 1E) and left box (Figure 1G), indicating the absence of natural preference and difference in the samples. In addition, no significant difference (*p* > 0.05) was found between the two groups of mice in the right box (Figure 1F) and left box (Figure 1H) after drug administration, indicating no differences in drug administration or addiction degree.

### 3.2. Disturbance of Intestinal Microbiota in Mice Induced by MA before and after Exercise

The alpha diversity among the three groups of mice (Chao, and observed_species) was not remarkably different from each other. The alpha diversity of group C was significantly different from that of group Ea (Shannon *p* = 0.0450 and Simpson *p* = 0.0370), and the alpha diversity of group Ma was significantly different from that of group Ea (Shannon *p* = 0.0378 and Simpson *p* = 0.0171). Figure 2A demonstrates the PCoA results with substantial variability in the composition of the gut microbial community in the three groups of mice. Figure 2B demonstrates the differential microbiome in the gate, and the results indicated that in the MA-treated mice, *Desulfobacterota* was the only microbiome in the gate that remarkably increased after MA administration and then markedly decreased after exercise. Figure 2C–E demonstrate the differences in the genus, family, and ASV levels among the three groups, respectively. In particular, 33 genera, 17 families, and 130 classes of ASV differed among the three groups. Figure 3 demonstrates the dominant microbiota in each group of LEfSe, and the results revealed 22 dominant microbiota in group C, 29 in group Ma, and 23 in group Ea. The bacterial classifications that differed most among the three groups of mice were *Bacteroidota*, *Proteobacteria*, *Desulfobacterota*, *Campilobacterota*, and *Cyanobacteria*. The details of the changes can be found in Table 1.

### 3.3. Differential Expression of Amino Acid Levels in MA-Induced Mice before and after Exercise

Figure 4A shows the heat map of amino acid levels between the three groups of mice. Results of the study showed that leucine (*p* [Ma/C] = 0.04, *p* [Ea/Ma] = 0.02), lysine (*p* [Ma/C] = 0.04, *p* [Ea/Ma] = 0.04), proline (*p* [Ma/C] = 0.02, *p* [Ea/Ma] = 0.01), tyrosine (*p* [Ma/C] = 0.04, *p* [Ea/Ma] = 0.05), and valine (*p* [Ma/C] = 0.04, *p* [Ea/Ma] = 0.02) increased after MA administration and decreased after exercise intervention. Asparagine (*p* = 0.03) and glutamine (*p* = 0.04) increased after MA administration, whereas alanine (*p* = 0.01), glycine (*p* = 0.02), isoleucine (*p* = 0.03), methionine (*p* = 0.05), and phenylalanine (*p* = 0.02) decreased after the exercise intervention. Details are shown in Table 2.

### 3.4. Differential Expression of SCFAs Levels in MA-Induced Mice before and after Exercise

Figure 4B–D show that the SCFAs had opposite trends after MA administration and locomotor intervention. Results showed that only hexanoic acid (*p* [Ma/C] = 0.01, *p* [Ea/Ma] = 0.03; Figure 4A), isovaleric acid (*p* [Ma/C] = 0.01, *p* [Ea/Ma] = 0.00; Figure 4B), and pentanoic acid (*p* [Ma/C] = 0.00, *p* [Ea/Ma] = 0.00; Figure 4C) showed opposite trends, whereas acetic acid (*p* [Ma/C] = 0.00, *p* [Ea/Ma] = 0.00) had the same trend after MA administration and exercise intervention. Butyric acid, isobutyric acid, and propionic acid showed no significant differences between groups Ma and C (*p* > 0.05), but those in group Ea were lower than those in group Ma (*p* < 0.05). Details are shown in Table 3.

### 3.5. Relationships of Gut Microbiota with Amino Acids and SCFAs

In the present study, we selected differentials that showed opposite trends between groups Ma and C and groups Ma and Ea for analyses. Figure 5A shows the ranking of the differential microbiome, SCFAs, and amino acids between MA addiction and exercise. Results showed that *Oscillibacter*, *Alloprevotella*, *Colidextribacter*, isovaleric acid, *Faecalibaculum*, and pentanoic acid ranked the top six in importance, among which seven categories of the intestinal microbiome, two SCFAs, and one amino acid were in the top 10 ranked in importance.

Moreover, a correlation analysis between some differential substances was performed. Figure 5B shows the correlation matrix between the differential substances. *{Eubacterium}_ventriosum*_group (*r* = 0.69); valine was positively correlated with *Ruminococcus* (*r* = 0.64) and RF39 (*r* = 0.84); leucine was positively correlated with *Ruminococcus* (*r* = 0.6) and RF39 (*r* = 0.88); and tyrosine (*r* = 0.51), proline (*r* = 0.69), and lysine (*r* = 0.5) were positively correlated with *Ruminococcus*. Pentanoic acid was negatively correlated with *Oscillibacter* (*r* = −0.5), and proline was negatively correlated with *Peptococcus* (*r* = −0.5).

## 4. Discussion

The effect of exercise on MA abuse has been verified by multiple sources. It can prevent MA-induced oxidative stress and blood–brain barrier disruption in brain microvessels [52] and also prevent the occurrence of MA-induced neurological abnormalities [53]. However, the effect of exercise on the gut microbiota and its metabolite levels in hosts after MA abuse is unknown. Therefore, in this study, after 7 days of CPP training, the mice spent considerably more time in the companion box than in the non-companion box to develop an addiction to the drug and establish the MA model. Subsequently, the levels of gut microbiota, amino acids, and SCFAs were assessed in each of the three groups of mice after 2 weeks of exercise intervention/regular feeding. The results showed that the levels of gut microbiota, amino acids, and SCFAs were considerably different in the three groups of mice. Aerobic exercise can modify to some extent the changes in microbiota structure, amino acids, and SCFA levels induced by MA administration in the intestine.

Effect of aerobic exercise on the intestinal microbiota of MA mice. In this study, 9 microbiotas were significantly reduced and 14 microbiotas were significantly increased in the Ma group compared with the C group; 9 microbiotas were significantly reduced and 12 microbiotas were significantly increased in the Ea group compared with the Ma group; 14 of the differences had opposite intestinal microbiota trends, and 7 had the same intestinal microbiota trends. The harmful bacterium *Ruminococcus* predicts serum 5-hydroxytryptamine and brain N-acetylaspartate levels [54] and is associated with the abuse of various psychoactive substances [55,56,57]. In the current study, *Ruminococcus* was found to be in a state of decline after the exercise intervention. However, the neutral bacterium *Prevotella* was positively associated with mucin synthesis, which increases intestinal permeability and is an important condition for MA, leading to pro-inflammatory effects [24]. Several studies have shown that exercise can reduce the abundance of such bacteria and maintain intestinal health [58,59,60]. The current study also found that *Alloprevotella* rose after MA administration and fell after exercise intervention. Surprisingly, most beneficial bacteria, such as *Muribaculaceae*, *{Eubacterium}_ventriosum_group*, and *Faecalibaculum*, were considerably elevated in MA and decreased substantially after the exercise. Most of these beneficial bacteria are extremely associated with mood or cognitive dysfunction [61,62,63], and growing evidence shows that depression and anxiety are the most predominant comorbidities of MUD [64] and that impaired cognitive function as well as changes in decision-making, are also important manifestations of MUD [65]. Therefore, whether aerobic exercise can improve MA-induced mood and depressive disorders by modulating such microbiomes is worth considering.

Effect of aerobic exercise on intestinal amino acid levels in MA mice. Seven amino acids were remarkably increased in the Ma group compared with the C group; 10 amino acids were considerably decreased in the Ea group compared with the Ma group; and five of the differential amino acids had opposite trends. Abundant evidence showed that excitatory amino acids asparagine and glutamic acid, as extremely important neurotransmitters in the central nervous system, are involved in the excitotoxicity of dopamine neurons in MUD [66]. The current study also found that asparagine and glutamine (a product of glutamic acid) were markedly elevated in the Ma group. Aerobic exercise has been reported to increase the utilization of excitatory amino acids, decrease the accumulation of excitatory amino acids in the body, and reduce the toxic effects of excitatory amino acids. We speculated that aerobic exercise led to a remarkable decrease in both amino acids in the Ea group [67]. Leucine and valine can signal to mTORC1 to drive intracellular signaling cascade responses [68,69], mTORC1-mediated signaling in the nucleus ambiguously directs the development of behavioral sensitivity by MA [70]. Running wheel exercise with 60% resistance could increase the sensitivity of mTORC1 to leucine [71]. However, after the start of the aerobic exercise, it becomes more dependent on the catabolism of essential amino acids, such as leucine and valine, for replenishment due to low glycogen stores [72]. The organism itself is relatively deficient [73], and in this case, leucine and valine would be in a state of decline after exercise without external supplementation, which would be consistent with the results of the current study. However, in the case of the results of the current study, the presence of this result may reduce the behavioral sensitivity due to MA administration.

The effect of aerobic exercise on intestinal SCFAs in MA mice. Four SCFAs were remarkably increased in the Ma group mice compared with the C group; six SCFAs were substantially decreased, and one SCFA was considerably increased in the Ea group compared with the Ma group; three differential SCFAs had opposite trends, and one differential SCFA had the same trend. Elevated levels of hexanoic acid and isovaleric acid have been reported to be associated with depression [74]. Both amino acids were markedly elevated after MA administration and may be an important condition that causes depression in patients with MUD. Unfortunately, the results of this paper cannot establish a causal relationship. Pentanoic acid reduces synaptic dysfunction and behavioral deficits due to MA-induced neuroinflammation by inducing IL-10 production [75,76]. Studies have shown that increased physical activity leads to lower SCFA concentrations in older patients with insomnia [77], and this paper found that hexanoic acid, isovaleric acid, and pentanoic acid all began to decline as well after 2 weeks of moderate-intensity aerobic exercise, which is consistent with the results of the previous study. However, more studies have shown that higher aerobic capacity corresponds to a higher SFCA production rate [78]. However, studies on the relationships of exercise with hexanoic acid, isovaleric acid, and pentanoic acid have been rare. Finally, how SCFAs change under conditions of moderate-intensity aerobic exercise after MA administration is worth considering.

We performed random forest ranking and correlation analysis to clarify the link between them. Preliminary random forest results revealed that the top 10 differentials ranked as markers of the opposite trend between the aerobic Ea group and the no-intervention Ma group after MA administration could be divided into three main categories according to their effects: 1. related to anxiety and depression and other emotional disorders: *Muribaculaceae*, proline, and isovaleric acid; 2. related to memory and other cognitive functions and other disorders: *Oscillibacter*, *Faecalibaculum*, *Negativibacillus*, proline, and pentanoic acid; 3. others: *Alloprevotella*, *Colidextribacter*, and *Uncultured*. Immediately thereafter, we selected the presence of oppositely trending microbiomes, amino acids, and SCFAs among the three groups and performed a correlation analysis. The correlations between different microbiomes and different amino acids showed that *Ruminococcus* and *RF39* were positively correlated with valine, leucine, and tyrosine; *Negativibacillus* was negatively correlated with proline and lysine; *Oscillibacter* was negatively correlated with pentanoic acid and tyrosine; and *Peptococcus* was negatively correlated with proline. The correlation of differential amino acids with differential SCFAs showed that proline, lysine, valine, and leucine were positively correlated with pentanoic acid. When we linked gut microbiome, amino acids, and SCFAs, we were surprised to find that *Negativibacillus*, which is associated with cognitive dysfunction, such as learning memory, was negatively correlated with proline, whereas proline was positively correlated with pentanoic acid. The results suggest that changes in the microbiota and its metabolites may be associated with cognitive dysfunction in MUD. Improving cognitive dysfunction in MUD by modulating the microbiota with exercise is an important direction for the future, and we expect to test this hypothesis.

## 5. Conclusions

In our previous study, we selected 2 weeks of moderate-intensity aerobic exercise intervention in group Ea [47]. We found 23 dominant microbiota, 5 amino acids, and 3 SCFAs that were considerably higher in group Ma than in group Ea. The top 10 markers with opposite trends between groups Ea and Ma after MA administration were *Oscillibacter*, *Alloprevotella*, *Colidextribacter*, *Faecalibaculum*, *Uncultured*, *Muribaculaceae*, *Negativibacillus*, *Negativibacillus*, proline, isovaleric acid, and pentanoic acid. *Negativibacillus* and proline, which are associated with cognitive dysfunction, such as reduced learning memory, were negatively correlated, whereas proline was positively correlated with pentanoic acid. The results suggest that aerobic exercise can reverse the disruption of gut microbiota and changes in amino acid and SCFA levels induced by MA administration. In addition, aerobic exercise, which reduces pentanoic acid levels by increasing the abundance of *Negativibacillus* and lowering proline, can be considered for the treatment of MA-induced cognitive dysfunction. The results suggest that aerobic exercise improves the disturbance of the gut microbiota and that its metabolite changes induced by MA administration may be related to cognitive function.

## Figures and Tables

**Figure 1 metabolites-13-00361-f001:**
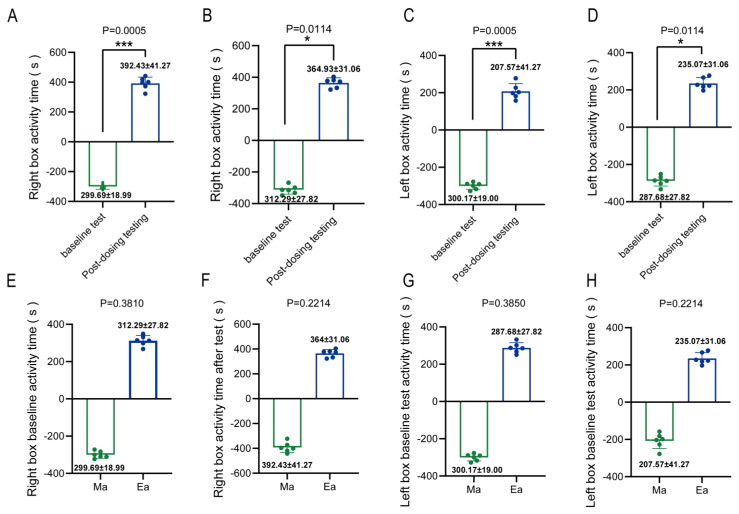
Addictive behavior of Methamphetamine-administered mice, setting the right box as the companion box. (**A**) Right box activity time of mice in group Ma; (**B**) Right box activity time of mice in group Ea; (**C**) Left box activity time of mice in group Ma; (**D**) Left box activity time of mice in group Ea; (**E**) Right box baseline activity time of mice in group Ma versus Ea; (**F**) Right box test activity time of mice in group Ma versus Ea; (**G**) Left box baseline activity time of mice in group Ma versus Ea; (**H**) Left box test activity time of mice in group Ma versus Ea. * Indicates post-dosing versus baseline level, *p* < 0.05; *** indicates post-dosing versus baseline level, *p* < 0.001.

**Figure 2 metabolites-13-00361-f002:**
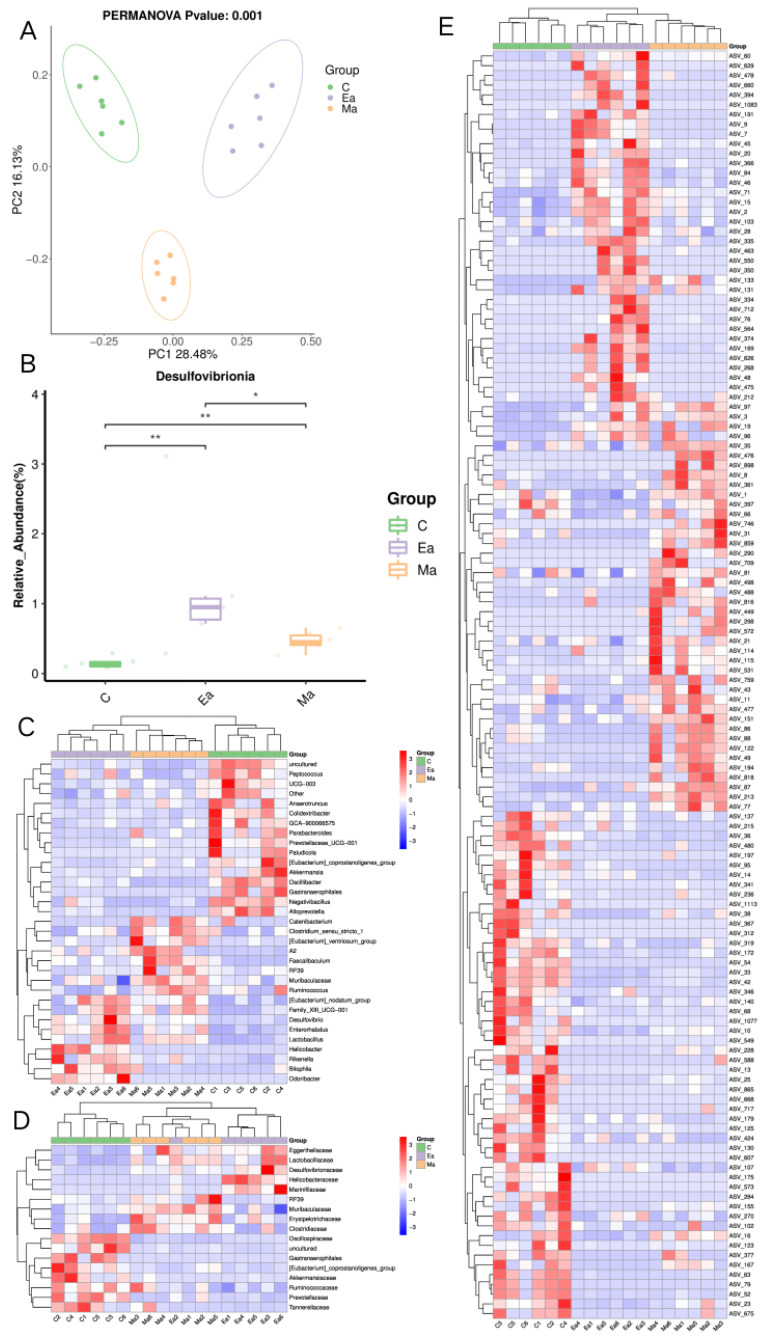
Changes among the intestinal microbiota of the three groups of mice. (**A**) PCoA showed significant differences in the ASV classification of the intestinal microbiome of the three groups of mice. (**B**) There was a significant change in the mean and standard deviation of the phylum Desulfobacterota in the three groups. (**C**) 33 bacterial genera. (**D**). 17 bacterial families. (**E**). 130 ASVs differed among the three groups. For heat map panels, it is all supervised clustering. * represents the two compared, *p* < 0.05; ** denotes the difference between the two, *p* < 0.01.

**Figure 3 metabolites-13-00361-f003:**
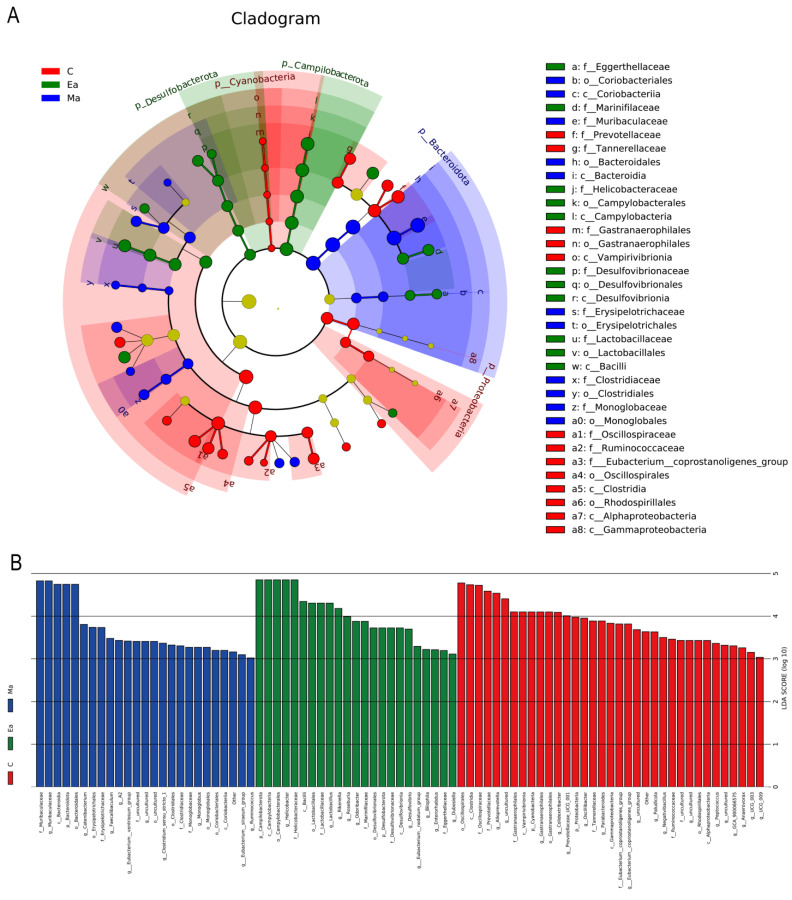
Dominant bacteria taxa in the different groups were identified using LEfSe. (**A**) Annotated branching diagram of differential microbiota; (**B**) Differential microbiota score diagram of differential species. A total of 74 microbial groups with statistically significant and biologically consistent differences were identified. The microbial groups with the greatest differences in the three groups include *Bacteroidota* and *Proteobacteria.* In (**A**,**B**): “red” indicates group C, “blue” indicates group Ma, and “green” indicates group Ea.

**Figure 4 metabolites-13-00361-f004:**
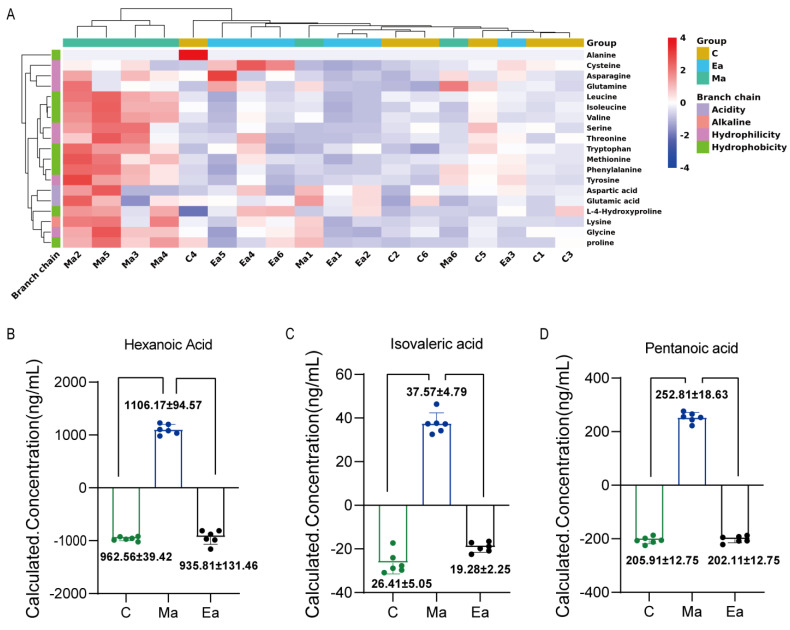
Comparison of concentration levels of intestinal metabolites SCFAs and amino acids among three groups of mice. (**A**) Changes in the levels of 19 amino acids in the intestine of three groups of mice; (**B**) Hexanoic acid content of mice in each group; (**C**) Isovaleric acid content of mice in each group; (**D**) Pentanoic acid content of mice in each group. * Indicates group Ma compared with group C, *p* < 0.05, ** indicates group Ma compared with group C, *p* < 0.01; # indicates group Ma compared with group C, *p* < 0.05; ## indicates group Ma compared with group C, *p* < 0.01.

**Figure 5 metabolites-13-00361-f005:**
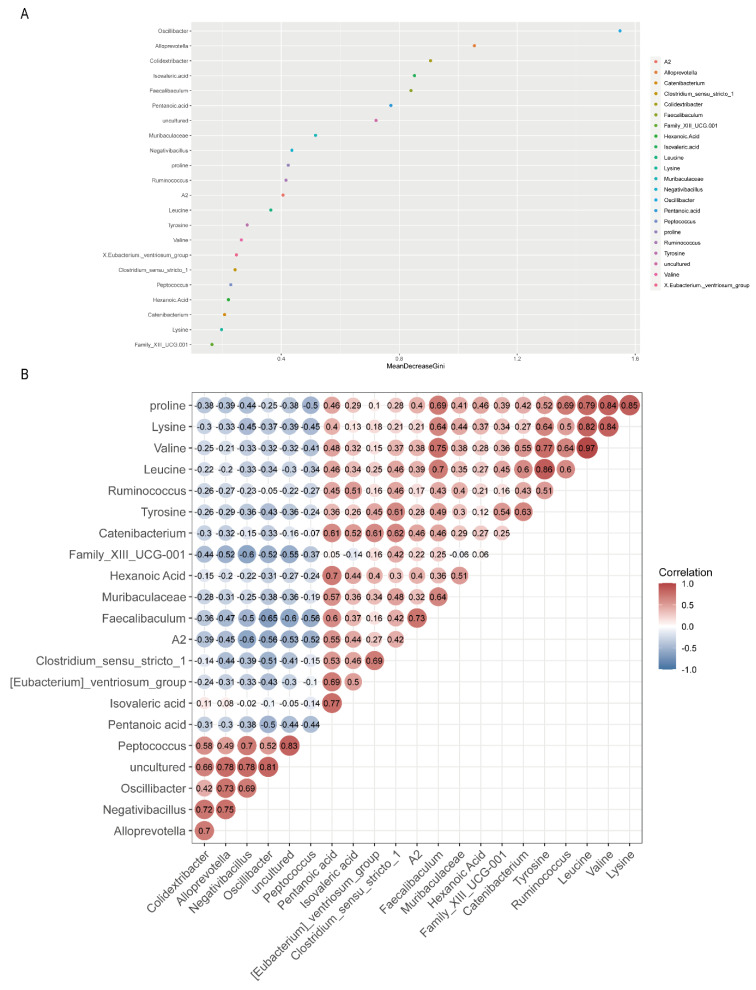
The relationship between different microbiota and SCFA and neurotransmitters. (**A**) Random Forest ranking of the differential microbiome, SCFAs, and amino acids; (**B**) Heat map of correlation between the differential microbiome, SCFAs, and amino acids. Related methods for generating graphs using Pearson correlation calculations. The redder the color, the more positive the correlation; the bluer the color, the more negative the correlation.

**Table 1 metabolites-13-00361-t001:** Table of the relationship between the changes in genus levels among the three groups of mice.

After MA Administration (Ma/C)		After Exercise Intervention (Ea/Ma)		
Taxon	Trend	Taxon	Trend	*p*-Value
*Oscillibacter*	↓	*Oscillibacter*	↑	0.0000021
*uncultured*	↓	*uncultured*	↑	0.0000055
*Negativibacillus*	↓	*Negativibacillus*	↑	0.0000478
*Alloprevotella*	↓	*Alloprevotella*	↑	0.0002397
*Faecalibaculum*	↑	*Faecalibaculum*	↓	0.0008663
*Enterorhabdus*	↑	*Enterorhabdus*	↑	0.0008799
*Colidextribacter*	↓	*Colidextribacter*	↑	0.0028403
*Gastranaerophilales*	↓			0.0029036
*Bilophila*	↑	*Bilophila*	↑	0.0039085
*Odoribacter*	↑	*Odoribacter*	↑	0.0044914
*Clostridium_sensu_stricto_1*	↑	*Clostridium_sensu_stricto_1*	↓	0.005689
*GCA-900066575*	↓	*GCA-900066575*	↓	0.0066066
*Lactobacillus*	↑	*Lactobacillus*	↑	0.0070059
*Catenibacterium*	↑	*Catenibacterium*	↓	0.0073016
*Muribaculaceae*	↑	*Muribaculaceae*	↓	0.0102365
*Peptococcus*	↓	*Peptococcus*	↑	0.0108253
*Ruminococcus*	↑	*Ruminococcus*	↓	0.013363
*{Eubacterium}_ventriosum_group*	↑	*{Eubacterium}_ventriosum_group*	↓	0.0135516
*{Eubacterium}_nodatum_group*	↑	*{Eubacterium}_nodatum_group*	↑	0.0150138
*A2*	↑	*A2*	↓	0.0151286
*Desulfovibrio*	↑	*Desulfovibrio*	↑	0.0269891
*Paludicola*	↓			0.0353126
*Family_XIII_UCG-001*	↑	*Family_XIII_UCG-001*	↓	0.0452011

Note: ‘↑’ indicates that the expression level of the former is higher than that of the latter; ‘↓’ means that the expression level of the two is lower than that of the latter.

**Table 2 metabolites-13-00361-t002:** The relationship of amino acid level changes between the three groups of mice.

After MA Administration (Ma/C)			After Exercise Intervention (Ea/Ma)		
Amino Acid	Trend	*p*-Value (Ma/C)	Amino Acid	Trend	*p*-Value (Ea/Ma)
Asparagine	↑	0.03	Alanine	↓	0.01
Glutamine	↑	0.04	Glycine	↓	0.02
Leucine	↑	0.04	Isoleucine	↓	0.03
Lysine	↑	0.04	Leucine	↓	0.02
proline	↑	0.02	Lysine	↓	0.04
Tyrosine	↑	0.04	Methionine	↓	0.05
Valine	↑	0.04	Phenylalanine	↓	0.02
-	-	-	proline	↓	0.01
-	-	-	Tyrosine	↓	0.05
-	-	-	Valine	↓	0.02

Note: ‘↑’ indicates that the expression level of the former is higher than that of the latter; ‘↓’ means that the expression level of the two is lower than that of the latter.

**Table 3 metabolites-13-00361-t003:** The relationship between the changes in SCFAs levels among the three groups of mice.

After MA Administration (Ma/C)			After Exercise Intervention (Ea/Ma)		
SCFAs	Trend	*p*-Value (Ma/C)	SCFAs	Trend	*p*-Value (Ea/Ma)
Acetic Acid	↑	0.00	Acetic Acid	↑	0.00
Hexanoic Acid	↑	0.0117	Butyric Acid	↓	0.0444
Isovaleric acid	↑	0.0010	Hexanoic Acid	↓	0.0296
Pentanoic acid	↑	0.0007	Isobutyric acid	↓	0.0013
-	-	-	Isovaleric acid	↓	0.0032
-	-	-	Pentanoic acid	↓	0.0004
-	-	-	Propionic acid	↓	0.0299

Note: ‘↑’ indicates that the expression level of the former is higher than that of the latter; ‘↓’ means that the expression level of the two is lower than that of the latter.

## Data Availability

The datasets used during the current study are not publicly available due to participant confidentiality issues. They are available to corresponding authors upon reasonable request. The results of the data in this study can be found on “Baidu.com” (https://pan.baidu.com/s/1dkziTAMwm4qeXktM1LAmHg) with extraction code 1234.

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
