# Peer review of "Effect of Aerobic Exercise on Intestinal Microbiota with Amino Acids and Short-Chain Fatty Acids in Methamphetamine-Induced Mice"

_metabolites, 2023, doi:10.3390/metabo13030361_

Round 1
Reviewer 1 Report
Major concerns:
The manuscript by Liang Xin et al. reports about the changes in gut microbiota composition, SCFAs and amino acid level in mice after methamphetamine administration and exercise intervention. Unfortunately, the manuscript is not acceptable for publication in its current state and must be improved in terms of clarity, including grammar corrections.
Specific comments:
Through the text the word “flora” must be corrected into “microbiome”.
The Latin names of bacterial genera and families must be written in italics (through the text and in the figures and tables).
Abstract
Replace “Methamphetamine” with “methamphetamine”. There are also other words that must be corrected on the same way.
Line 35 - the keyword “microbiota-gut–brain axis” - This was not investigated and must be excluded. Namely, effect on brain was not examined.
Line 360 – “Relationship between amino acids and gut microbiota classification for different SCFAs” - The title is not logical and must be corrected.
Line 488 – “This result suggests that the "microbiota-gut-brain axis" may be an effective way to treat cognitive dysfunction in MUD”- Must be reformulated, because the influence of obtained results on the "microbiota-gut-brain axis" was not examined.
Line 494 – “The effect of exercise on the microbiota–gut–brain axis” - It was not investigated and must be corrected. Namely, effect on brain was not examined.
Line 509 – The part of sentence “by modulating the microbiota gut–brain axi.” must be excluded because the influence of obtained results on the "microbiota-gut-brain axis" was not examined.
Figures and tables of the manuscript - The titles of figures and tables must contain the full names of the abbreviations used.
Figure 2. (A) “Changes in the intestinal microbiota of mice” – It is not clear which parameter is used (filum, genus, family)?
Figure 3. (A) “Annotated branching diagram of differential species” – microbial species are not shown on the figure. The title must be corrected.
Figure 3. (B) “Score diagram of differential species. A total of 74 microbial groups with statistically significant and biologically consistent differences were identified.” – microbial species are not shown on the figure. The title must be corrected. The results must be shown on the way that each category of the bacteria (filum, genus, family) must be shown separately and the text must be readable (bigger letters). Reorganising of the presentation of the results is necessary.
Figure 4. (A) “19 amino acids were detected, of which leucine, lysine, proline, tyrosine, and valine were significantly different after dosing and after exercise” - this comment about results must be excluded from the title
Figure 5. “Importance ranking of differential metabolites and the relationship between them” - The title must be corrected. Namely, microbiota (wrongly named flora) is not a metabolite.
Table 1 “genus level change table” – The title needs to be corrected because it is not understandable without reading the text of the manuscript
Table 2 “amino acid level change table” - The title needs to be corrected because it is not understandable without reading the text of the manuscript
Table 3 “SCFAs level change table” - The title needs to be corrected because it is not understandable without reading the text of the manuscript
Author Response
Dear Editors and Reviewers:
We thank the editors for processing our manuscript entitled "Effect of Aerobic Exercise on Changes in the Intestinal Microbiota with Amino Acids and Short-chain Fatty Acids in Methamphetamine-induced Mice" (Manuscript ID: metabolites-2186050). We also thank the reviewers for their comments and observations on this manuscript. These comments are valuable and helpful for us to revise and improve the paper, and they are also important guidance for our research.
We have revised this manuscript with the comments of the two reviewers, and we hope to receive your approval. The revised parts are marked in red in the manuscript.
The main changes in the paper and the responses to the reviewers' comments are as follows:
Reviewer 1:
Specific comments:
Point 1: Through the text, the word “flora” must be corrected into “microbiome”.
Response 1: We sincerely thank you for your valuable comments on this study.
We have made changes to the manuscript.
Point 2: The Latin names of bacterial genera and families must be written in italics (through the text and in the figures and tables).
Response 2: Thanks for your valuable comments.
We have made changes to the manuscript.
Abstract
Point 3: Replace “Methamphetamine” with “methamphetamine”. Other words must be corrected in the same way.
Response 3: Thanks for your valuable comments.
We have made changes to the manuscript.
Point 4: Line 35- the keyword “microbiota–gut–brain axis” -This was not investigated and must be excluded. Namely, the effect on the brain was not examined.
Response 4: Thanks for your valuable comments.
We have made changes to the manuscript.
Point 5: Line 360 – “Relationship between amino acids and gut microbiota classification for different SCFAs” -The title is not logical and must be corrected.
Response 5: Thanks for your valuable comments.
We have made changes to the manuscript.
Point 6: Line 488– “This result suggests that the "microbiota-gut-brain axis" may be an effective way to treat cognitive dysfunction in MUD”- Must be reformulated because the influence of obtained results on the "microbiota-gut-brain axis" was not examined.
Response 6: Thanks for your correction. This opinion is very helpful to us.
We have made changes to the manuscript.
Point 7: Line 494 – “The effect of exercise on the microbiota–gut–brain axis”-It was not investigated and must be corrected. Namely, the effect on the brain was not examined.
Response 7: We sincerely thank you for your valuable comments on this study.
We have made changes to the manuscript.
Point 8: Line 509– The part of the sentence “by modulating the microbiota gut–brain axis.” must be excluded because the influence of obtained results on the "microbiota-gut-brain axis" was not examined.
Response 8: Thank you for your valuable comments.
We have made changes to the manuscript.
Point 9: Figure 2. (A) “Changes in the intestinal microbiota of mice” – It is not clear which parameter is used (film, genus, family).
Response 9: Thank you very much for your correction.
We have made changes to the manuscript.
Point 10: Figure 3. (A) “Annotated branching diagram of differential species” – microbial species are not shown in the figure. The title must be corrected.
Response 10: Thank you for your precious comments and advice.
We have made changes to the manuscript.
Point 11: Figure 3. (B) “Score diagram of differential species. A total of 74 microbial groups with statistically significant and biologically consistent differences were identified.” – microbial species are not shown in the figure. The title must be corrected. The results must be shown in the way that each category of the bacteria (filum, genus, family) must be shown separately and the text must be readable (bigger letters). Reorganizing the presentation of the results is necessary.
Response 11: Thanks for your valuable comments.
We are sorry, but we have used vector images that can be enlarged to ensure that the lettering is as clear as possible when enlarged. However, due to typographical issues, we are temporarily unable to give the 3B chart a larger image of the letters in the manuscript. We will provide vector images of all individual images separately for checking.
Point 12: Figure 4. (A)“19 amino acids were detected, of which leucine, lysine, proline, tyrosine, and valine were significantly different after dosing and after exercise”- this comment about results must be excluded from the title
Response 12: Thanks for your valuable comments.
We have made changes to the manuscript.
Point 13: Figure 5. “Importance ranking of differential metabolites and the relationship between them” - The title must be corrected. Namely, microbiota (wrongly named flora) is not a metabolite.
Response 13: Many thanks to the reviewers for their comments and suggestions.
We have made changes to the manuscript.
Point 14: Table 1“genus level change table” –The title needs to be corrected because it is not understandable without reading the text of the manuscript
Response 14: We sincerely thank you for your valuable comments on this study.
We have made changes to the manuscript.
Point 15: Table 2“amino acid level change table” - The title needs to be corrected because it is not understandable without reading the text of the manuscript
Response 15: Thank you for your valuable comments.
We have made changes to the manuscript.
Table 16: “SCFAs level change table” - The title needs to be corrected because it is not understandable without reading the text of the manuscript
Response 16: Thank you very much for your correction. We have made changes to the manuscript.

Reviewer 2 Report
Comments and Suggestions for Authors
The manuscript written by Liang et al is an original article that aimed to highlight the effect of aerobic exercise on changes of amino acids (AAs) and short-chain fatty acids (SCFAs) in methamphetamine-induced mice. The manuscript is well written in terms of the English language. The topic is interesting for researchers, providing sufficient background to understand its message, with the experimental and analysis part well described and thus reproducible. However, it requires improvement by reviewing a few major and minor issues:
Major:
The discussions are speculative, and require reformulation or new approaches, as follows:
- In the discussions chapter, the findings of certain AAs and SCFAs changes in mice as an effect of aerobic exercises were discussed in the context of the literature but the authors did not explain them in a way that brings clarification to the subject. As such, the authors should address specifically the following issues: Why are only certain AAs and SCFAs modified as a result of physical effort and others not? What is the cause of these changes? What is the usefulness of these changes and possibly their clinical applicability in the future?
- The authors are quite certain of their findings, although the study was conducted on 3 groups of 6 mice each. It is quite difficult to validate certain theories on these lots. Moreover, the study does not address its weaknesses at all!!!!
- There are many bold statements that should either be rephrased, explained in more detail, or supplemented by studies that support them:
a) Lines 398-401: if your results showed that levels of gut microbiota, AAs, and SCFAs were different in the 3 groups, you cannot imply that “…aerobic exercise can reverse to some extent the imbalance in microbiota homeostasis...” without defining what is the normal status of the microbiota in terms of the composition and proportions between species or the metabolites they produce.
b) Line 412: if you speculate on the result, please define in general which are the beneficial or harmful bacteria.
c) Lines 442-446: I don't understand the explanation in depth. The muscle under effort first uses glycogen reserves to provide glucose (glycogenolysis), followed by gluconeogenesis from AAs. You mentioned Leucine and Valine. I agree with Valine being a glucogenic AA, but Leucine is a ketogenic AA. As such, why would Leucine decrease after exercise?
- Lines 505-507: this statement is not supported by the study data. You did not evaluate the cognitive dysfunction in your study, so the statement “…can be considered for the treatment of MA-induced cognitive dysfunction” is not in accordance.
Minor:
- Starting with the abstract and throughout the entire article, all intestinal flora species must be written in italics.
- Line 32: Please specify what type of effort it is (duration, intensity). I mean low, medium, or high?
- Line 77: “on” is missing after “…may have effects”
- Lines 110-111: “…were elucidated” is a bold statement Please rephrase.
- There are abbreviations that have not been defined: line 116 SPF, line 133 CPP, and line 313 ASVs.
After the amendment of the above comments in the manuscript, I would be in favor of publishing the article.

Author Response
Dear Editors and Reviewers:
We thank the editors for processing our manuscript entitled "Effect of Aerobic Exercise on Changes in the Intestinal Microbiota with Amino Acids and Short-chain Fatty Acids in Methamphetamine-induced Mice" (Manuscript ID: metabolites-2186050). We also thank the reviewers for their comments and observations on this manuscript. These comments are valuable and helpful for us to revise and improve the paper, and they are also important guidance for our research.
We have revised this manuscript with the comments of the two reviewers, and we hope to receive your approval. The revised parts are marked in red in the manuscript.
The main changes in the paper and the responses to the reviewers' comments are as follows:
Reviewer 2:
Major:
The discussions are speculative, and require reformulation or new approaches, as follows:
Point 1: In the discussions chapter, the findings of certain AAs and SCFAs changes in mice as an effect of aerobic exercises were discussed in the context of the literature but the authors did not explain them in a way that brings clarification to the subject. As such, the authors should address specifically the following issues: Why are only certain AAs and SCFAs modified as a result of physical effort and others not? What is the cause of these changes? What is the usefulness of these changes and possibly their clinical applicability in the future?
Response 1: We sincerely thank you for your valuable comments on this study.
We have added to the incomplete description of the AA section. However, the SCFAs part of its description does not present the hypothesis reason is that existing studies on the specific relationship between exercise and Hexanoic Acid, isovaleric acid, and Pentanoic acid are scarce, and almost no reference literature can be found in recent years. Therefore, we have described the results, functions, possible effects on the MUD, and future research directions. We hope we can get your understanding and support.
Point 2:The authors are quite certain of their findings, although the study was conducted on 3 groups of 6 mice each. It is quite difficult to validate certain theories on these lots. Moreover, the study does not address its weaknesses at all!!!!
Response 2: Thank you for your valuable comments.
This experiment mainly refers to some literature, which can ensure the feasibility and accuracy of the experiment. We appreciate and understand the reviewers' comments and will consider further improving future experiments based on the reviewers' comments.
We attach here the DOI information of the references:DOI: 10.1016/j.bbr.2020.112971、DOI: 10.1111/adb.12502、DOI: 10.1016/s0306-4603(03)00082-0
Many bold statements should either be rephrased, explained in more detail, or supplemented by studies that support them:
Point 3: a) Lines 398-401: if your results showed that levels of gut microbiota, AAs, and SCFAs were different in the 3 groups, you cannot imply that “…aerobic exercise can reverse to some extent the imbalance in microbiota homeostasis...” without defining what is the normal status of the microbiota in terms of the composition and proportions between species or the metabolites they produce. a)
Response 3: Thank you very much for your correction.
We have made changes to the manuscript.
Point 4: b) Line 412: if you speculate on the result, please define in general which are the beneficial or harmful bacteria. b)
Response 4: Thank you for your precious comments and advice.
We have made changes to the manuscript.
Point 5:c) Lines 442-446: I don't understand the explanation in depth. The muscle under effort first uses glycogen reserves to provide glucose (glycogenolysis), followed by gluconeogenesis from AAs. You mentioned Leucine and Valine. I agree with Valine being a glucogenic AA, but Leucine is a ketogenic AA. As such, why would Leucine decrease after exercise? c)
Response 5: Thanks for your valuable comments.
Leucine is one of the three essential amino acids that increase muscle mass and help muscle recovery after workouts and is the only amino acid that can replace glucose during fasting. Leucine is extremely easy to convert into glucose and regulates blood sugar better than isoleucine and valine. These functions make it useful when the body is under stress. It is converted to glucose for energy in the exercise state.
Point 6:Lines 505-507: this statement is not supported by the study data. You did not evaluate the cognitive dysfunction in your study, so the statement “…can be considered for the treatment of MA-induced cognitive dysfunction” is not in accordance.
Response 6: Thanks for your correction. This opinion is very helpful to us.
We have made changes to the manuscript.
Minor:
Point 7: Starting with the abstract and throughout the entire article, all intestinal flora species must be written in italics.
Response 7: We sincerely thank you for your valuable comments on this study.
We have made changes to the manuscript.
Point 8: Line 32: Please specify what type of effort it is (duration, intensity). I mean low, medium, or high?
Response 8: Thank you for your valuable comments.
We have made changes to the manuscript.
Point 9: Line 77: “on” is missing after “…may have effects”
Response 9: Thank you very much for your correction.
We have made changes to the manuscript.
Point 10: Lines 110-111: “…were elucidated” is a bold statement Please rephrase.
Response 10: Thank you for your precious comments and advice.
We have made changes to the manuscript.
Point 11: Some abbreviations have not been defined: line 116 SPF, line 133 CPP, and line 313 ASVs.
Response 11: Thanks for your valuable comments.
We have made changes to the manuscript.

Round 2
Reviewer 2 Report
I appreciate the effort of the authors to respond to the comments raised and to carry out most of the changes suggested by me.
I believe that after correcting the remaining small editing mistakes (which would be the job of the proofreading team), the article can be published.
I would mention just one small objection: in your answer to me you said something extremely wrong: "Leucine is extremely easy to convert into glucose...". Fortunately, this does not appear in the manuscript, because Leucine is one of the 2 amino acids that do NOT form glucose, but ketone bodies.
Author Response
Responses to editors and reviewers
Dear editors and reviewers.
We thank the editors for processing our manuscript entitled Effect of Aerobic Exercise on Changes in the Intestinal Microbiota with Amino Acids and Short-chain Fatty Acids in Methamphetamine-induced Mice (Manuscript ID: metabolites-2186050). We also thank the reviewers for their comments and observations on this manuscript. These comments were valuable for us to revise and improve the paper, and they also provided important guidance for our research.
After careful study of these comments. For this reason, we have revised the manuscript in accordance with the reviewers' comments and hope that we can obtain your approval. The revised sections are marked in the manuscript with red font.
The major revisions of the paper and the responses to the reviewers' comments are as follows.
Point 1: Review by a native English speaker (for appropriate syntax and agreement); there are still multiple agreement, syntax, and capitalization/lowercase issues throughout the manuscript that are not appropriate for publication.
Response 1: We sincerely thank you for your valuable comments on this study.
This issue has been revised by the author in the manuscript.
Point 2: a. Title: Please capitalize “chain” and “induced”
Response 2: Thanks for your valuable comments.
This issue has been revised by the author in the manuscript.
Point 3: b. Line 266: please specify versions of Qiime2 and R that were used in your analyses.
Response 3: Thanks for your valuable comments.
This issue has been revised by the author in the manuscript.
Point 4: c. Line 267: Please specify the distance/scaling parameters used for your PcoA visualization/analysis (i.e. bray-curtis, jaccard, weighted or unweighted unifrac, etc). Please additionally specify the parameters used to generate your heatmap (scaling- Euclidean, ward, etc distances and were unsupervised or supervised clustering algorithms used?)
Response 4: Thanks for your valuable comments.
We have made changes to the manuscript.
Point 5:d. Line 267: Please spell out what lefse stands for here (versus in line 273), as this is the first place where the acronym shows up. Second, what were your lefse thresholds of an LDA score used to define an ASV as a “dominant” contributor to a treatment group (i.e. was an LDA score of 2 or higher, etc used to identify LDA’s that distinguished one treatment from another?)
Response 5: Thanks for your valuable comments.
We have made changes to the manuscript.
Point 6:e. Lines 270-275 would read best coming directly after where line 267 is currently. The typical flow of analysis/results in microbiome studies is alpha diversity, beta diversity, then taxonomy and differential abundance analyses.
Response 6: Thanks for your correction. This opinion is very helpful to us.
We have made changes to the manuscript.
Point 7:e1. Lines 276-281: Please state the parameters used to generate random forest analysis (i.e. number of permutations, any other assumptions going into test). Please ensure that the two software/website links you note for the HIPLOT and random forest analysis are included in your references bibliography and as in-text citations here. Finally, please specify if statistical hypothesis testing was performed to evaluate significance of the correlations you calculated between metabolite abundances. and gut microbiota ASV’s.
Response 7: We sincerely thank you for your valuable comments on this study.
We have made changes to the manuscript.
Point 8: f. Figure 1- Do the dot plot bars indicate standard error or standard deviation? Please specify. Also, please make the dot plot bars two-way (ie. Versus having the bar only extend upwards along the y axis of each graph bar).
Response 8: Thank you for your valuable comments.
We have made changes to the manuscript.
Point 9:g. Figure 2- In the legend, please specify what parameter the ellipses in panel A (pcoa) specify- i.e. are they reflecting the 95% confidence interval of each group, the range, etc? For heat map panels, please specify in the legend if the heat map is supervised versus unsupervised clustering. For panel B, please explain in the legend what spread is shown (are these median and ranges or mean and standard error, etc box plots?) If there is any significant differences between these treatments on these panels, please indicate this on the panel and state it in figure legend.
Response 9: Thank you very much for your correction.
We have made changes to the manuscript.
Point 10:g2. Figure 3- In the legend, please include the distance/scaling parameters used to generate the dendrogram in A and spell out the name of each treatment group that corresponds to each color.
Response 10: Thank you for your precious comments and advice.
We have made changes to the manuscript.
Point 11:h. Figure 4- Do the dot plot bars indicate standard error or standard deviation? Please specify. Also, please make the dot plot bars two-way (ie. Versus having the bar only extend upwards along the y axis of each graph bar).
Response 11: Thanks for your valuable comments.
We have made changes to the manuscript.
Point 12:Table 3- Please include more significant digits to the p-values so that values do not reach p = 0.00. In cases where p is so small that it is reported as 0, you can consider reporting it as p < ### (where the # is the lowest value that your hypothesis test can detect, such as p < 1.0 E-13 if your test detected differences down to 13 significant figures).
Response 12: Thanks for your valuable comments.
We have made changes to the manuscript.
Point 13: Figure 5- In the legend, please include the correlation method used to generate the figure (Spearman’s, Pearson, etc). Please indicate what different colors of the circles with the correlation values indicate.
Response 13: Many thanks to the reviewers for their comments and suggestions.
We have made changes to the manuscript.
